# Imaging Cancer-Associated Fibroblasts (CAFs) with FAPi PET

**DOI:** 10.3390/biomedicines10030523

**Published:** 2022-02-23

**Authors:** Laura Gilardi, Lighea Simona Airò Farulla, Emre Demirci, Ilaria Clerici, Emanuela Omodeo Salè, Francesco Ceci

**Affiliations:** 1Division of Nuclear Medicine, IEO European Institute of Oncology IRCCS, 20141 Milan, Italy; laura.gilardi@ieo.it (L.G.); ligheairo@gmail.com (L.S.A.F.); 2Department of Oncology and Hemato-Oncology, University of Milan, 20122 Milan, Italy; 3Department of Nuclear Medicine, Faculty of Medicine, Yeditepe University, Istanbul 34718, Turkey; emredemirci@gmail.com; 4Division of Pharmacy, IEO European Institute of Oncology IRCCS, 20141 Milan, Italy; ilaria.clerici@ieo.it (I.C.); eomodeo@ieo.it (E.O.S.)

**Keywords:** cancer-associated fibroblast, fibroblast activation protein, FAPi, PET/CT, theranostics

## Abstract

The tumor microenvironment (TME) surrounding tumor cells is a complex and highly dynamic system that promotes tumorigenesis. Cancer-associated fibroblasts (CAFs) are key elements in TME playing a pivotal role in cancer cells’ proliferation and metastatic spreading. Considering the high expression of the fibroblast activation protein (FAP) on the cell membrane, CAFs emerged as appealing TME targets, namely for molecular imaging, leading to a pan-tumoral approach. Therefore, FAP inhibitors (FAPis) have recently been developed for PET imaging and radioligand therapy, exploring the clinical application in different tumor sub-types. The present review aimed to describe recent developments regarding radiolabeled FAP inhibitors and evaluate the possible translation of this pan-tumoral approach in clinical practice. At present, the application of FAPi-PET has been explored mainly in single-center studies, generally performed in small and heterogeneous cohorts of oncological patients. However, preliminary results were promising, in particular in low FDG-avid tumors, such as primary liver and gastro-entero-pancreatic cancer, or in regions with an unfavorable tumor-to-background ratio at FDG-PET/CT (i.e., brain), and in radiotherapy planning of head and neck tumors. Further promising results have been obtained in the detection of peritoneal carcinomatosis, especially in ovarian and gastric cancer. Data regarding the theranostics approach are still limited at present, and definitive conclusions about its efficacy cannot be drawn at present. Nevertheless, the use of FAPi-based radio-ligand to treat the TME has been evaluated in first-in-human studies and appears feasible. Although the pan-tumoral approach in molecular imaging showed promising results, its real impact in day-to-day clinical practice has yet to be confirmed, and multi-center prospective studies powered for efficacy are needed.

## 1. Introduction

### 1.1. Targeting the Tumor Microenvironment

Tumors are composed of two interdependent compartments: the malignant cells and the stroma, or tumor microenvironment (TME), which may account for up to 90% of the mass in common malignancy, such as breast, stomach, and pancreatic carcinomas. It is a highly dynamic and heterogeneous system composed of immune cells, fibroblasts, precursor cells, endothelial cells, signaling molecules, and extracellular matrix (ECM) components, which interact closely with tumor cells, contributing to tumorigenesis [1].

The natural progression of TMEs can be described in this sequence [2]:

(1)A small cluster of homogeneous cancer cells. These tumors evade immune surveillance as they are in a very early stage of development or because they are a newly metastasized colony.(2)Tumors with lymphocyte infiltration that release cytokines and directly engage with cancer cells, recruiting blood cells to the tumor. At the same time, nearby macrophages and fibroblasts are converted into tumor-associated macrophages (TAMs) and cancer-associated fibroblasts (CAFs).(3)Tumors without infiltrating lymphocytes, encapsulated by CAFs with ECMs. They are filled with many stromal cells, including TAMs, CAFs, and myeloid-derived suppressor cells (MDSCs) and do not release cancer cells into the blood circulation.(4)Tumors with a subgroup of cancer cells that undergo epithelial-mesenchymal transition, which downregulate some genes (such as E-cadherin, β-catenin, cytokeratin 5 and 6) and upregulate other genes (such as E-cadherin, vimentin, Snail, Slug, Twist, ZEB1 and 2, S100A4, MMP2 and 3, α-smooth muscle actin), so these tumors become metastasized, activating mobility-enhancing genes (such as S100 CBPP) and releasing cancer cells into the blood circulation that are often chaperoned by stromal cells.

Accordingly, key elements in the tumor stroma are CAFs: metabolically active spindle-shaped cells with enhanced proliferative and migratory properties that release many growth factors and proinflammatory cytokines, such as transforming growth factor-β (TGF-β), vascular endothelial growth factor (VEGF), and interleukin-6 (IL-6). The causes driving the transformation of fibroblasts to CAFs are not entirely understood. However, the transformation is driven by the occurrence of mutations, including inactivation of TP53 and PTEN [3] and loss of heterozygosity (LOH) [4].

Through this biochemical crosstalk with the surrounding cells of TME, CAFs play a pivotal role in cancer cell invasion, migration and growth, metabolic reprogramming, immunosuppression, and angiogenesis [5]. A distinguishing feature of CAFs is their high expression of fibroblast activation protein (FAP), a type II membrane-bound glycoprotein belonging to the dipeptidyl peptidase 4 (DPP4) family. FAP has both dipeptidyl peptidase and endopeptidase activity, has a large extracellular domain, and is associated with the regulation of the extracellular matrix [6] (Figure 1). This integral protein is coexpressed with DPP4 in the alpha cells of Langerhans islets, in multipotent bone marrow stromal cells, and is also slightly present in the cervix and uterine stroma (during the proliferative cycle) [7,8,9]. In contrast to CAFs, normal fibroblasts have no or very slight FAP expression and, therefore, FAP expression is low in normal adult human tissues. Like CAFs, FAP expression is instead increased in activated fibroblasts in the case of tissue damage, remodeling, or inflammation, and, therefore, in benign conditions, such as wound healing, arthritides, and myocardial infarction [10,11,12].

### 1.2. Developing PET Radiopharmaceuticals

The differential expression of the protein in normal tissue compared with tumors/inflammation makes FAP a promising target for molecular imaging of a large variety of tumors and for some non-oncological diseases. For this purpose, FAP-targeting radiopharmaceuticals based on FAP-specific inhibitors (FAPis), such as ^68^Ga-FAPi-02 and ^68^Ga-FAPi-04, have recently been developed [13,14]. These quinoline-based radiotracers bind to the enzymatic domain of FAP with very high specificity, and the complex is then rapidly internalized. Biodistribution studies on tumor-bearing mice and on mixed population of different cancers showed high intratumoral uptake of the tracers and fast renal clearance, with very low uptake in normal organs (especially the brain, oral mucosa, and liver), leading to a higher tumor-to-background ratio (TBR) and improving the diagnostic performance of PET imaging [13,15,16].

The use of DOTA or other chelators offers the possibility of easily incorporating therapeutic isotopes, such as ^177^Lu or ^90^Y, in these compounds, allowing a theranostic approach [14]. Nevertheless, the therapeutic application of FAPi tracers is still impaired by their relatively short tumor retention time, even if FAPi-04 already obtained an improvement in tumor retention compared to FAPi-02 (75% washout) with a 50% of tumor uptake from 1 to 3 h after injection. Lately, FAPi-46 was developed, which allows a theranostic approach due to its longer tumor residence time [17].

Attempts were initially made to label FAPi with covalently attached ^18^F and favorable results were obtained with NOTA-containing FAPi-74 [18]. Indeed, due to the longer half-life of ^18^F and lower positron energy than ^68^Ga (half-life of 110′ and positron energy of 0.65-MeV versus 68′ and 1.90-MeV), labeling FAPi with ^18^F would reduce the costs of production in case of on-site cyclotron and facilitate its distribution together with improved spatial resolution due to the lower positron energy [19]. In addition, local on-demand production in centers already equipped with a ^68^Ge/^68^Ga generator would also be simplified by using NOTA chelator, allowing chelation with ^68^Ga at room temperature. [18]. Conversely, the use of ^68^Ge/^68^Ga generators in clinical practice is impaired by the size of the generator and thus by the maximum number of patients that can be handled per-synthesis and by the increasing prices of generators.

FAPi radiopharmaceuticals demonstrate intense radioactivity in the urinary tract, with the kidneys being the main excretory organs. Uptake of the radioactivity was also observed in the gallbladder and common bile duct, implying elimination via the hepatobiliary system as well. Moderate uptake of radioactivity was observed in other organs, including the submandibular gland, thyroid, and pancreas. Only minimal or mild physiological uptake was observed in other organs and tissue, including brain, parotid, oral mucosa, lung, myocardium, liver, intestine, fat, spine, and muscle. Differences in the biodistribution of [^18^F]FAPi-42 and [^68^Ga]Ga-FAPi-04 in normal organs might be due to the different lipophilicity of the NOTA-chelator and DOTA-chelator groups. This might influence the detection of lesions in specific regions, especially for pancreatic, gallbladder, and biliary tract tumors [20]. Drug-related pharmacologic effects or physiologic responses have so farnever been observed in clinical studies, implying the use of FAP inhibitors for PET imaging. The radiopharmaceuticals’ injection was well tolerated, and no side effects have been described.

### 1.3. Article Selection and Data Extraction

A comprehensive search was performed using the Ovid platform and a comparison of the Embase and Medline databases. No time restrictions were applied. The following search strategy was used: (“cancer associated fibroblast” OR “cancer-associated fibroblast” OR “CAF” OR “CAFs”) AND (“fibroblast activation protein” OR “fibroblast activation protein inhibitor OR “FAPi” OR “FAP”) AND (“Positron Emission Tomography” OR “PET”). The web search was implemented with a manual search (authors consultation and web-search included articles). Only studies in English were selected. The literature search was updated until 10 December 2021. From all studies, we selected for this review the most relevant articles, evaluating manuscripts reporting about the use of radiopharmaceuticals for PET imaging targeting cancer-associated fibroblasts with fibroblast activation protein inhibitors, based on the following criteria: (1) original article or case series or case reports in the (2) English language regarding the (3) use of PET other FAPi (either ^68^Ga or ^18^F). Two authors (LG and LSAF) independently reviewed the abstracts and titles of the retrieved studies for inclusion in the review based on the inclusion criteria and removed duplicates. A second review was performed to delete additional studies outside the scope of this review. Disagreements were resolved through consultation with a third author (FC) or consensus. The authors tabulated and organized relevant studies and performed a comprehensive qualitative narrative synthesis of both tabulated studies and non-tabulated articles.

## 2. Clinical Application of Fapi Pet in Oncology

### 2.1. Preclinical Studies

Early preclinical studies included the use of FAPI-01 and FAPI-02 radiolabeled with ^125^I and ^68^Ga/^177^Lu, respectively. The [^125^I]FAPI-01 was then no longer included in preclinical studies because of its time-dependent efflux and enzymatic deiodination, although it showed rapid internalization in both human embryonic kidney (HEK) cells transfected with murine FAP and human fibrosarcoma HT-1080 cells transfected with FAP in vitro. In contrast, [^68^Ga]FAPI-02 revealed increased binding and uptake in human cells expressing FAP in vitro and in vivo due to its increased stability. Biodistribution studies of [^177^Lu]FAPI-02 showed maximal uptake at 2 h after administration in mice carrying human FAP-transfected HT-1080 tumors. In biodistribution studies on tumor-bearing mice and on the first cancer patients, the authors also found high intratumoral uptake of the tracer and fast body clearance, resulting in high contrast images and negligible exposure of healthy tissue to radiation [16]. 

Similar results have emerged from another preclinical study, which demonstrated that the contrast in imaging can be further improved by performing chemical modifications on the structure of FAPI, thereby improving the binding and pharmacokinetics of FAP in most derivatives. The authors also proved how higher doses of radioactivity can be administered while minimizing damage to healthy tissue, thereby improving the therapeutic outcome [17].

At this point, the need to improve the lipophilicity and prolong the tumor retention time emerged. After synthesizing 15 novel quinoline-based radiotracers, the authors analyzed in vivo pharmacokinetic studies by small-animal PET imaging on FAP-transfected HT-1080 xenografts. The five most promising ^68^Ga-labeled tracers (FAPI-21, FAPI-35, FAPI-36, FAPI-46, and FAPI-55) showed low background activity, rapid tumor accumulation, and prevalent renal clearance. FAPI-36 was excluded because of its prolonged systemic circulation, whereas FAPI-21, FAPI-35, and FAPI-55 showed increased uptake in hepatic and muscle tissues compared with FAPI-04. From the above, it appears that the most promising derivative in this series is FAPI-46 because of its good tumor accumulation and its highest tumor-to-background ratios [17].

### 2.2. Pan-Tumoral Radiotracer

Different tumor types might share common driver mutations: this assumption underlies the concept of pan-cancer analysis [21]. FAP-targeting radiopharmaceuticals might be potentially superior to ^18^F-FDG and thus there is growing interest in its application as a pan-tumoral radiotracer. This approach inspired a translational prospective exploratory study [22] evaluating the role of FAPi-PET as a pan-cancer imaging biomarker for FAP expression in 141 patients with 14 different types of cancer (bile duct, bladder, breast, esophagus, colon, liver, stomach, lung, ovary, uterus, oropharynx, prostate, pancreas, and kidney). All patients were eligible for surgery and, thus, immunohistochemical confirmation of FAP expression was obtained in all cases. The authors found that FAPi-PET was positive in more than 50% of cases from 11 cancer types, with variability in the intensity of FAP expression strongly associated with FAP expression assessed by immunohistochemistry in the surgery specimen. FAP expression was higher in cancers of the bile duct, bladder, colon, esophagus, stomach, lung, oropharynx, ovary, and pancreas; average in breast and uterus cancer; and liver, prostate, and renal cell cancer showed only low FAP expression. In addition, all four evaluated metastatic lesions were FAPi-avid, which has important implications in theranostics. Although this study showed how FAPi could play a central role in the context of pan-cancer analysis, the limited number of patients enrolled for each tumor sub-type affected the study’s reproducibility and, thus, broader consideration cannot be drawn at this stage.

A similar conclusion was obtained in a previous study reporting [15] about FAPi-PET uptake in 80 patients and 28 different tumor types. The primary aim was to assess if FAPi-PET might improve tumor delineation (e.g., for radiotherapy planning) in patients with inconclusive or clinically unsatisfactory standard of care imaging. The highest uptake was detected in patients with sarcoma, esophageal, breast, cholangiocarcinoma, and lung cancer, whereas pheochromocytoma, renal cell, differentiated thyroid, and gastric cancers were the lowest. Furthermore, the authors also found that despite the high intratumoral and interindividual variability, the low background activity, resulting in high tumor-to-background ratios, leads to excellent image contrast (Table 1).

### 2.3. Gliomas, Primary Liver Cancer, and Gastro-Entero-Pancreatic Cancers

First experiences in mixed populations demonstrated that FAPi, unlike FDG, had a significant lower background distribution in the brain, liver, and oral/pharyngeal mucosa and very low unspecific gastric and intestinal/peritoneal uptake [13,15]. These characteristics make FAPi a very promising radiopharmaceutical for the detection of moderate-to-low FDG-avid tumors and for the evaluation of primary and secondary lesions in regions with low TBR at FDG-PET/CT, such as the brain and liver. In their pilot study [23], Röhrich and colleagues characterized the uptake of FAP ligands in 18 glioma patients, 13 with isocitrate dehydrogenase (IDH)-wildtype glioblastoma WHO grade IV and 5 with IDH-mutant glioma. The authors observed elevated tracer uptake in IDH-wildtype glioblastomas and WHO grade III/IV IDH-mutant astrocytomas but not in WHO grade II astrocytomas. Therefore, if these findings are confirmed in larger populations, FAPi-based PET/CT may be useful for non-invasive characterization of gliomas and of their malignant progression from grade II to higher tumor grades. Moreover, the same 13 patients with IDH-wildtype glioblastomas were evaluated for radiotherapy planning [24]. FAPi-PET-based gross tumor volumes (GTVs) were incongruent with MRI GTVs. Indeed, increases in GTVs were highly significant for all FAP-specific PET thresholds. The clinical and therapeutic impact of these results has yet to be addressed.

Primary hepatobiliary tumors showed different patterns of FDG uptake. Hepatocellular carcinoma (HCC), the most frequent primary tumor of the liver, is characterized by an FDG uptake equal to the normal liver tissue, leading to a high false-negative rate in FDG-PET detection. Due to its favorable distribution characteristics, FAPi could overcome these performance deficiencies.

Indeed, recent studies demonstrated higher FAPi uptake in HCC and intrahepatic cholangiocarcinoma (ICC) compared to FDG, with a significantly higher sensitivity [25]. Moreover, FAPi-PET/CT proved to have an equivalent sensitivity for the detection of primary tumors compared to contrast-enhanced CT and MRI and a significantly higher detection rate than FDG-PET for all malignant lesions, including extrahepatic disease [26]. Guo et al. also demonstrated increased ^68^Ga-FAPI-04 uptake in the liver parenchyma of patients with cirrhosis, with a significantly higher FAPi-TBR in patients without cirrhosis [26]. Further studies are needed to evaluate the impact of these results on the diagnostic performance of FAPi-PET, in terms of the definition of the correct extension of the disease and the detection of small intrahepatic lesions. FAPi represents an ideal radiopharmaceutical to detect liver metastasis as well, due to the lower physiological uptake, leading to potential upstaging and exact identification of tumor locations. In addition, targeted and personalized treatments, such as liver radio-guided surgery, image-guided radiotherapy, or radio-embolization, might be proposed based on the information derived by FAPi-PET. However, several benign conditions, including hepatic cirrhosis, showed FAPi uptake, thus limiting the role of this radiotracer in primary liver cancer, since the presence of hepatic cirrhosis is a common risk factor, especially in patients affected by HCC.

Gastroenteropancreatic (GEP) tumors often disseminate to the liver. In this setting, FAPi-PET was found to be superior to FDG-PET in the detection of liver metastases in small studies including patients with gastric, colorectal, and other GEP tumors [27,28]. Besides, FAPi-PET demonstrated a higher detection rate for primary GEP tumors compared to FDG-PET, with higher TBR and clearer tumor identification [28], considering the lower physiological gastric or bowel uptake. In a retrospective bicentric study focused on gastric cancer staging, FAPi-PET was superior to FDG-PET for the detection of the primary tumor, with a sensitivity of 100% vs. 82%. In particular, the radiopharmaceutical FAPi outperformed FDG in signet ring cell carcinoma (7/7 vs. 4/7), probably due to the low expression level of glucose transporter 1 in this histological type. SUVmax of T2-4 tumors was significantly higher than SUVmax of T1 tumors in FAPi-PET, highlighting the possibility of a non-invasive evaluation of the infiltration degree in primary gastric cancers [29] (Table 2). These preliminary and encouraging results show that, in the future, FAPi-PET may play an important role in brain, liver and GEP cancers, outperforming FDG, and need to be validated in prospective clinical trials on larger samples. 

### 2.4. Head and Neck Cancers

Accurate tumor staging is crucial in head and neck cancer for making an adequate treatment choice, namely the identification of contralateral metastasis. Despite the use of contrast-enhanced CT and MRI, radiological imaging often fails to correctly identify the extension of the disease, due to the diffuse tumor infiltration in complex structures and to the presence of underlying and concomitant inflammatory processes. FDG-PET is affected by several limitations as well, mainly related to high physiological uptake in healthy tissue, such as salivary glands, brain, and oral cavity, and inflammatory uptake in cervical muscles or lymph nodes. Moreover, the presence of FDG-avid brown adipose tissue may impair image reading in FDG-PET scans. In this scenario, FAPi-PET could provide a potential solution, as highlighted by the first single-center studies performed in a cohort of nasopharyngeal carcinoma patients [30,31].

In both papers, FAPI-PET proved to be superior to FDG for exact T staging. Low tracer uptake in normal structures adjacent to the tumor allowed imaging of primary tumors with higher TBR and better lesion delineation. Skull base and intracranial invasion are clearly visualized on ^68^Ga-FAPI PET due to the low brain tissue uptake. Further studies [32,33] confirmed that FAPi-PET led to significant upstaging of the disease compared to FDG-PET, with a consequent change in the therapy management. However, some concerns are still related to the definition of positive lymphnode since inflammatory nodes might show low FAP overexpression (Table 3). 

**Table 1 biomedicines-10-00523-t001:** Oncological setting: Pan-tumoral radiotracer.

Authors	*n* of Patients	Tumor Type	Clinical Setting	Injected Activity	Acquisition Timing	Image Analyses	Reference Standard	^68^Ga-FAPI Performance	Highest FAPi Uptake	Lowest FAPi Uptake
Mona CE et al. [22]	141	Various cancer (14 types)	Biodistribution and kinetics	174–185 MBq	54–96 min	S	HP	SE 80.9%	Bile duct, bladder, colon, esophagus, stomach, lung, oropharynx, ovary and pancreas cancer	Liver, prostate, and renal cell cancer
Kratoch wil C et al. [15]	80	Various cancer (28 types)	Staging, Restaging, RT planning	122–312 MBq	60 min	S	HP, imaging follow-up	Low uptake (≤6): 7/28 TT; Mean uptake (6 > SUV < 12): 14/28 TT; High Uptake (≥12): 7/28 TT	Lung, breast and esophageal cancer, cholangiocellular carcinoma and sarcoma (SUVmax ≥ 12)	Pheochromocytoma, renal cell, differentiated thyroid, adenoid cystic and gastric cancer (SUVmax ≤ 6)
Chen H et al. [32]	68	Various cancer (13 types)	Staging, Restaging	1.8–2.2 MBq/Kg	60 min	V, S	HP, imaging and clinical follow-up	T: SE 86.4%	T: liver, gastric, pancreatic and cervical cancer	T: oroesophageal and lung cancer
Chen H et al. [33]	75	Various cancer (12 types)	Staging, Restaging	1.8–2.2 MBq/Kg	60 min	V, S	HP	T: SE 98.2%	Pancreatic, liver and oroesophageal cancers, sarcoma and cholangiocarcinoma (SUVmax ≥ 12)	Brain cancer
								N: SE 86.4%, SP 58.8%		
								M: SE 83.8%, SP 41.7%		

V, visual analyses; S, semi-quantitative analyses; HP, histopathology; T, primary tumor; N, lymph node(s); M, distant metastases; TT, tumor types; SE, sensitivity; SP, specificity.

**Table 2 biomedicines-10-00523-t002:** Oncological setting: Gliomas, primary liver cancer and gastro-entero-pancreatic cancers.

Authors	*n* of Patients	Tumor Type	Clinical Setting	Injected Activity	Acquisition Timing	Image Analyses	Reference Standard	^68^Ga-FAPI Performance	^18^F-FDG Performance	MRI Performance
Röhrich M et al. [23]	18	Gliomas	Staging, Restaging	150–250 MBq	30 min (FA-Pi04); 10, 60 and 180 min (FAPi-02)	S	MRI	SE 83.3%	-	SE 100%
Windisch P et al. [24]	13	GBM	RT planning	150–250 MBq	30 min	S	MRI	SE 100%	-	SE 100%
Guo W et al. [26]	34	Hepatic nodules	Staging	148–259 MBq	60 min	V, S	HP, imaging follow-up	SE 87.4%	SE 64.9%	-
Şahin E et al. [27]	31	GEP	Staging and follow-up after treatment	2–3 MBq/Kg	45 min	V, S	Imaging follow-up, tumor biomarker findings, HP	SE 93.5% (patient based)	SE 71% (patient based)	-
								SE 95.9% (lesion based)	SE 79.6% (lesion based)	-
Pang Y et al. [28]	35	GI tract	Staging, Restaging	1.8–2.2 MBq/Kg	60 min	V, S	HP	SE 100%	SE 43.8%	-
								T: SE 100%	T: SE 52.6%	-
								N: SE 78.6%, SP 82.1%	N: SE 53.6%, SP 89.3%	-
								M: SE 88.6%, SP 28.6%	M: SE 57.1%, SP 85.7%	-
Jiang D et al. [29]	38	Gastric cancer	Staging	111–185 MBq	60 min	S	HP	T: SE 100%	T: SE 75%	-
								N: SE 60%, SP 92.9%	N: SE 50%, SP 92.9%	-

V, visual analyses; S, semi-quantitative analyses; HP, histopathology; T, primary tumor; N, lymph node(s); M, distant metastases; SE, sensitivity; SP, specificity; GBM, glioblastoma; GEP, gastro-entero-pancreatic; GI tract, gastro-intestinal tract.

**Table 3 biomedicines-10-00523-t003:** Oncological setting: Head and neck cancers.

Authors	*n* of Patients	Tumor Type	Clinical Setting	Injected Activity	Acquisition Timing	Image Analyses	Reference Standard	^68^Ga-FAPI Performance	^18^F-FDG Performance	MRI Performance
Zhao L et al. [30]	45	Nasopharyngeal carcinoma	Staging, Restaging	1.8–2.2 MBq/Kg	40 min	V, S	HP, imaging follow-up	T: SE 86.7%	T: SE 84.4%	-
								N: SE 95%	N: SE 75.2%	N: SE 97.5%
Qin C et al. [31]	15	Nasopharyngeal carcinoma	Staging, Restaging	1.85–3.7 MBq/Kg	30–60 min	V, S	MRI	T: SE 100%	T: SE 100%	-
								N: SE 48%	N: SE 100%	-
								M: SE 100%	M: SE 0%	-
Chen H et al. [32]	68	Various cancer (13 types)	Staging, Restaging	1.8–2.2 MBq/Kg	60 min	V, S	HP, imaging and clinical follow-up	T: SE 86.4%	-	-
Chen H et al. [33]	75	Various cancer (12 types)	Staging, Restaging	1.8–2.2 MBq/Kg	60 min	V, S	HP	T: SE 98.2%	T: SE 82.1%	-
								N: SE 86.4%, SP 58.8%	N: SE 45.5%, SP 76.5%	-
								M: SE 83.8%, SP 41.7%	M: SE 59.5%, SP 58.3%	-

V, visual analyses; S, semi-quantitative analyses; HP, histopathology; T, primary tumor; N, lymph node(s); M, distant metastases; SE, sensitivity; SP, specificity.

Nevertheless, the main clinical application for FAPi-PET in head and neck cancer is the planning of image-guided radiotherapy, especially considering the high TBR. Recently, improved target volume delineation has been reported in comparison with contrast-enhanced CT and MRI [24,34]. This image-guided approach might have substantial implications in the planned target volume (PTV), probably leading to better regional control of the disease and less toxicity due to the more precise identification of the treatment field. Future studies are needed to define the optimal imaging time and thresholds of FAPi uptake for PTV delineation. A personalized radiotherapy approach can be created according to FAPi-PET images, with precise radiation dose escalation or de-escalation plans for tumor subvolumes or with plan adaptation due to microenvironment changes during treatment.

### 2.5. Breast Cancer

Breast cancer (BC) is a heterogeneous disease in terms of the pathological features, biological behaviors, therapeutic response, and prognosis. The tumors can be classified into subtypes distinguished by pervasive differences in their gene expression patterns and, consequently, phenotypes, where the key players are the estrogen receptor (ER), progesterone receptor (PR), and the human epidermal growth factor receptor 2 (HER2) [35,36]. The different statuses of receptor expression serve as biomarkers for the choice of specific and tailored treatment strategies. Although chemotherapy and targeted therapies have improved the survival of BC patients, many patients relapse or do not respond to first-line therapies. In the last years, it has become increasingly evident that breast cancer evolution is not solely dependent on the behavior of cancer cells, but also on the composition and biological function of TME and on the interactions between cancer cells and TME itself. CAFs represent the most abundant cell type of the BC microenvironment [37] and, consequently, FAP is an excellent candidate as an indirect tumor cell target.

To date, one prospective study assessed the role of FAPi-PET in patients with breast cancer, in comparison with FDG PET/CT [38]. The authors enrolled 20 patients with newly diagnosed or relapsed BC (15 and 5 patients, respectively). ^68^Ga-FAPI-04 PET/CT was superior to FDG-PET in detecting primary tumors, with a 100% and 95.6% sensitivity and specificity, respectively. Moreover, FAPi-PET detected more lymph node, hepatic, bone, and brain metastases due to the lower background activity and higher uptake in subcentimetric lesions. Future studies should focus on the role of FAPi-PET in diagnosis of different molecular and histological subtypes of BC (as FDG-PET has known limitations in low-grade hormone-positive tumors and in lobular carcinomas) and on its potential in the early detection of disease relapse and in the assessment of therapy response and the patient’s prognosis.

Another attractive application in the BC field concerns the use of FAPi-PET as a guide for the selection of patients for radio-ligand therapy (RLT) with FAP-specific inhibitors, using high-energy β-emitters, such as Lutetium-177. This theranostic approach could be of great value for patients with triple-negative tumors, where currently used agents targeting ER, PR, and HER2 are ineffective. 

### 2.6. Peritoneal Carcinomatosis

Peritoneal carcinomatosis (PC) is a complication regarding several malignancies and is generally associated with a poor outcome. Total peritonectomy and resection of the involved tissue with intraperitoneal chemotherapy is a primary treatment of PC with curative intention [39]. Two of the most important prognostic factors are the extent and volume of the PC. Thus, the pre-operative evaluation of possible peritoneal involvement is crucial and, currently, exploratory surgery is the gold standard for PC detection [40]. Conventional imaging (CT or MRI) is obviously a less invasive procedure compared to surgery. FDG-PET showed good diagnostic accuracy in detecting PC, even if is impaired by the spatial resolution limitation, and the pathological peritoneal thickness does not always show increased glycolytic metabolic activity [41]. Therefore, FAPi-PET might emerge as a leading diagnostic procedure in this specific setting. Recently, in a retrospective analysis, ^68^Ga-DOTA-FAPi-04 PET/CT was compared to FDG-PET for the evaluation of PC in 46 patients (13 with gastric cancer, 10 colorectal, 9 ovarian, 6 pancreatic, 2 lung, 2 appendiceal, 1 cervical, 1 endometrial, and 1 breast cancer and 1 primary PC). The authors observed a significant difference in the standard uptake values (SUV) of lesions, namely in PC from gastric cancer. FAPi-PET showed a higher peritoneal cancer index (PCI) and better sensitivity than FDG PET [42]. Likewise, a case report about a 63-year-old man who presented with rising carcinoembryonic antigen levels and unknown primary, and had both FDG-PET and FAPi-PET performed demonstrated that FAPi imaging is superior in the detection of primary gastric lesion (not FDG-avid) and a higher number of mesenteric nodules [43]. Another case report about a 60-year-old woman treated for BC showed that FAPI-PET detected more bone lesions and more favorable TBR for peritoneal nodules compared to FDG-PET [44]. Lastly, a case report of a 55-year-old woman, who underwent FDG-PET and FAPi-PET, showed higher accuracy for FAPi imaging in the correct identification of mesenteric and omentum nodules while both techniques correctly identified the primary pancreatic lesion. Follow-up FAPi-PET performed after three months was able to assess the response to cytoreductive surgery [45]. 

## 3. Radioligand Therapy Targeting the Tumor Micro-Environment

Theranostics is a neologism that merges the words therapeutics and diagnostic, defining the presence of a specific target that can be equally used both for PET imaging and radio-ligand therapy (RLT). Theranostics has been part of nuclear medicine for decades, starting from the use of radioactive iodine-131, to the application in neuroendocrine tumors labeling somatostatin analogues with ^68^Ga-, ^177^Lu-, or ^90^Y-labeled somatostatin analogs and, more recently, in prostate cancer targeting the prostate-specific membrane antigen (PSMA). The concept of “TME targeted RLT” is, at present, only a theoretical approach. Nevertheless, FAP inhibitors can be proposed as theranostics agents, as FAPi-ligands are chelator-based containing DOTA, which can be labeled by different isotopes. All solid tumors require stroma to grow beyond a minimal size of 1-2 mm and generate it, activating the host’s wound-healing response. Given the long-held notion of tumors as “wounds that do not heal” [46], and the increasing knowledge about the role of TME in tumorigenesis, the innovative approach of FAPi-based RLT consists in treatment of the supporting system that allows cancer cells to grow and reproduce. First in-human studies evaluated RLT with ^90^Y-FAPI-46, ^177^Lu-FAPI-46, and ^177^Lu-FAP-2286 on 9 [46], 18 [47], and 11 patients [48] with different cancers, respectively, and demonstrated the feasibility of this therapeutical approach. RLT was well tolerated, with acceptable side effects (predominantly hematological) and low radiation doses absorbed by non-target tissues, including the kidneys. For ^177^Lu-FAP-2286, the kidney-absorbed dose was comparable to that of ^177^Lu-PSMA-617 and to that delivered by ^177^Lu-DOTATATE with renal protection through amino-acid administration. The whole-body- and bone-marrow-absorbed dose were similar too [48]. The studies’ cohorts were too small and heterogeneous (and mostly comprising heavily pre-treated patients and/or metastatic high-burden cases). Thus, any reliable conclusion regarding the FAPi-based RLT efficacy cannot be drawn at this stage. At present, the clinical application of RLT targeting the TME is only theoretical, and dedicated studies powered by efficacy are needed, since different tumor subtypes responding to TME-targeted RLT, and the injected doses, types of radionuclides (α and/or β emitters), and the development FAP inhibitors with prolonged retention are under still investigation. 

## 4. Clinical Application of Fapi Pet in Non-Oncological Disease

FAP overexpression is observed in many healing processes and benign conditions, such as fibrosis, thus providing the opportunity for a hypothetical application of FAPi-PET even outside oncology. Fibroblasts play a pivotal role in cardiac tissue remodeling and wound healing [11] and the expression of FAP by activated cardiac fibroblasts increases after myocardial infarction and then declines over time [49]. A reduction in cardiac fibrosis and a restoration of systolic function have been achieved by ablation of FAP-positive cells in mice subjected to angiotensin II and phenylephrine [50]. Moreover, a retrospective study analyzing the scans of 229 patients showed an association between high FAPi signal intensities and the presence of a metabolic risk factor (such as arterial hypertension, diabetes mellitus, and obesity) and an increased focal uptake could be suggestive of an underlying cardiovascular disease [51].

FAPi-PET has also shown a role in IgG_4_-related diseases (IgG_4_-RD). A case report of a patient with IgG_4_-RD undergoing both FDG-PET and FAPi-PET showed that this quinoline-based radiotracer had an abnormal uptake in the uncinate process of the pancreas, in addition to sites indicated by FDG-PET/CT, except for some supra- and subdiaphragmatic lymph nodes [52]. Another similar case report demonstrated increased uptake in the lacrimal glands in FAPi-PET that was negative on FDG-PET but, as in the previous case, FDG-avid lymph nodes were negative on FAPi-PET [53].

## 5. Conclusions

Exploring the TME with molecular imaging is an attractive field of investigation and PET imaging with FAPi-PET is now gaining attention as “pan-tumoral” radiopharmaceuticals since CAFs are activated in many tumor subtypes. However, at present, information regarding the feasibility and efficacy of FAPi-PET is derived from single-center studies only, enrolling small and heterogenous cohorts of patients. The preliminary results are promising, as FAPi-PET seems to allow better evaluation of tumors with low FDG-avidity, leading to tumor upstaging through the detection of unknown distant metastases (especially in case of peritoneal carcinosis) and an improved target volume delineation for radiotherapy planning. Furthermore, FAPi-PET allows the in vivo visualization of TME, leading to a better comprehension of tumor heterogeneity, namely when performed together with FDG-PET. Finally, the first in-human studies highlighted the feasibility of radioligand therapy with FAPi inhibitors, targeting the TME in different tumor subtypes. Multi-center studies in larger cohorts, together with histological validation of FAPi-PET findings, are needed to assess the efficacy of FAPi-based imaging and to understand if, and when, the translation into clinical practice of this new promising approach would be feasible.

## Figures and Tables

**Figure 1 biomedicines-10-00523-f001:**
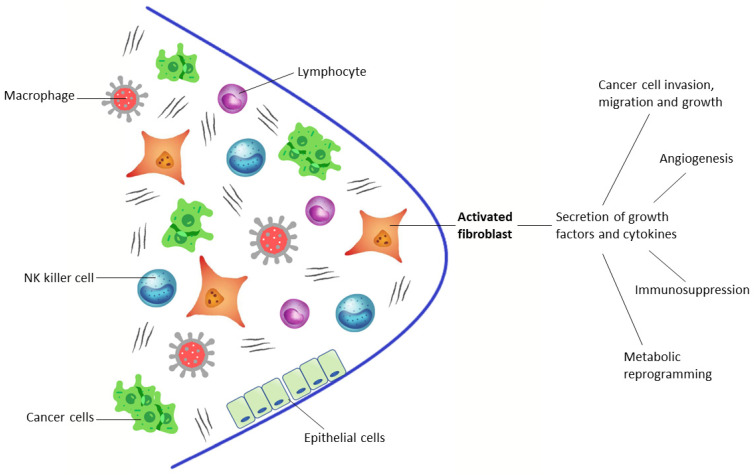
The tumor microenvironment consists of tumor cells and nonmalignant cells, such as lymphocytes, macrophages, NK cells, normal epithelial cells, and activated fibroblasts (CAFs).

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
