# Peer review of "Imaging Cancer-Associated Fibroblasts (CAFs) with FAPi PET"

_biomedicines, 2022, doi:10.3390/biomedicines10030523_

Round 1

Reviewer 1 Report

This is an interesting study reviewing recent developments on radiolabeled FAP inhibitors and their potential role in clinical practice. The manuscript is well-organized. In order to discuss the basic biological mechanisms, some pre-clinical animal research could be reviewed in added as a new subheading. 

Reviewer 2 Report

The author reviews the recent developments on radiolabeled FAP (fibroblast activation protein) inhibitors and the possible translation of this pan-tumoral approach in clinical practice. This paper shows a very well-organized work, the results from the available literature being comprehensively described. Therefore, I recommend its publication in Biomedicines.
